# A Eulerian Numerical Model to Predict the Enhancement Effect of the Gravity-Driven Motion Melting Process for Latent Thermal Energy Storage

**DOI:** 10.3390/e26020175

**Published:** 2024-02-19

**Authors:** Shen Tian, Bolun Tan, Yuchen Lin, Tieying Wang, Kaiyong Hu

**Affiliations:** Tianjin Key Laboratory of Refrigeration Technology, Tianjin University of Commerce, Tianjin 300134, China; 18340304408@163.com (B.T.); a1213141595@163.com (Y.L.); wangtieying@tjcu.edu.cn (T.W.); hky422@tjcu.edu.cn (K.H.)

**Keywords:** gravity-driven motion, melting process, phase-change material, melting rate enhancement, numerical modeling

## Abstract

Latent thermal energy storage (LTES) devices can efficiently store renewable energy in thermal form and guarantee a stable-temperature thermal energy supply. The gravity-driven motion melting (GDMM) process improves the overall melting rate for packaged phase-change material (PCM) by constructing an enhanced flow field in the liquid phase. However, due to the complex mechanisms involved in fluid–solid coupling and liquid–solid phase transition, numerical simulation studies that demonstrate physical details are necessary. In this study, a simplified numerical model based on the Eulerian method is proposed. We aimed to introduce a fluid deformation yield stress equation to the “solid phase” based on the Bingham fluid assumption. As a result, fluid–solid coupling and liquid–solid phase transition processes become continuously solvable. The proposed model is validated by the referenced experimental measurements. The enhanced performance of liquid-phase convection and the macroscopic settling of the “solid phase” are numerically analyzed. The results indicate that the enhanced liquid-phase fluidity allows for a stronger heat transfer process than natural convection for the pure liquid phase. The gravity-driven pressure difference is directly proportional to the vertical melting rate, which indicates the feasibility of controlling the pressure difference to improve the melting rate.

## 1. Introduction

The melting process is widely used in various fields utilizing thermal energy, such as solar photothermal conversion [1], electric fields [2] and aerospace engineering [3], due to the advantages of its stable temperature maintenance and efficient thermal energy conversion ability. One of the key features of this process is its inherent and considerable latent heat variation, which is emphasized in the development of thermal energy storage devices. This feature can reduce operating costs and device volume, thereby further increasing the flexibility and efficiency of thermal energy storage systems [4]. From the current trend of energy usage worldwide, it is forecast by the International Renewable Energy Agency (IRENA) that the global thermal energy storage market will triple in size by 2030, and the capacity of thermal storage devices will increase to over 800 GWh in the next 10 years [5]. Therefore, the in-depth combination of an efficient melting process with the development of thermal energy storage devices has the potential to meet urgent thermal energy storage needs.

The gravity-driven motion melting (GDMM) process, characterized by accelerated phase transitions, has attracted considerable research attention. It refers to the macroscopic relative motion between the liquid and the solid phase of a phase-change material (PCM) under gravity. This will cause a change in phase distribution so that the heated wall and the solid phase can be positioned close to, or apart from, each other. In most cases, extremely thin liquid films (in the order of hundreds of micrometers) can form between the heated wall and the solid phase and are associated with the close-contact melting (CCM) phenomenon [6]. However, it should be noted that GDMM is only one type of process in which the CCM phenomenon occurs. On the one hand, the fluidity of the liquid phase is effectively improved by macroscopic relative motion [7]. On the other hand, thin liquid films considerably reduce thermal resistance. Thus, it is reported that GDMM can shorten the charging time four-fold compared to the traditional melting process [8]. Therefore, it has become a potential method for application in various latent thermal energy storage (LTES) devices with packaged PCM containers [9]. It can also be combined with processes, such as PCM material optimization (using enhanced fluidity) and extended surface design as well as enhanced heat transfer methods with disturbance, to obtain more effective LTES devices [10,11,12].

The GDMM process simultaneously involves a fluid–solid coupling motion and liquid–solid phase transition. Simulations have been used in order to further investigate the physical details of this process. There are two main types of modeling methods: analytical and numerical. Analytical models describe a problem through overall process equations and can usually obtain highly accurate solutions. Recently, Aljaghtham et al. [13] established an analytical model for the coupled phase-change heat transfer between the Navier slip condition and energy transport. The influences of characteristic parameters on the transient transformation of the melting rate and liquid-film thickness were revealed. Rozenfeld et al. [14] proposed a theoretical model to study the CCM phenomenon in a horizontal casing tube with longitudinal finned inner tubes. This model considered the rotational motion of solids under gravity, primary melting on vertical fins with uneven temperature distribution, secondary melting on the shell, and frictional resistance on the shell. The results show that CCM significantly improved the heat transfer rate and shortened the melting time 2.5-fold.

Researchers can already predict local dynamic characteristic values, especially for thin liquid films, using analytical models. However, due to the impact of PCM packaging, more detailed numerical solutions are still needed to analyze the full parameters of the GDMM process. Therefore, numerical models considering both limited space and horizontally heated walls were proposed, mainly including simulations based on the Euler coordinate system. Interestingly, due to the fixed position of the mesh in the Euler coordinate system, most of these models use fluids to simulate the solid phase of the PCM. Pan et al. [15] established a numerical model of vertical cylindrical melting based on Fluent software. Although the model did not simulate the settling behavior of bulk solids, precise melting time and volume fraction results were obtained through modifications of the momentum equation. Kozak and Ziskind [16] established a numerical model based on the Euler coordinate system and enthalpy. This model can solve the problem of “solid phase” stretching and deformation by changing the state of the mesh at the top and bottom of the “solid phase”. This leads to the “solid phase” calculation domain changes being instantaneous, which seems similar to the modeling process of a Lagrangian coordinate system. Faden et al. [17] developed an enthalpy–porosity numerical model and added an extended Darcy term to the momentum equation to limit the deformation of the “solid phase”. An innovatively implicit algorithm was used to calculate the solid settling velocity. Their approach provided physically correct results for the vertical melting process but did not study other parameters such as the thickness of the liquid film and the impact intensity from side walls. Oskouei et al. [18] also used the enthalpy–porosity method to simulate the effects of CCM and natural convection on energy storage systems with annular fins. However, the enthalpy–porosity method is highly dependent on the value of the mushy zone constant. Adjustments must be made to the mushy zone constant, which may be difficult to achieve for each simulation case.

The difficulty in solving the GDMM process based on a Euler coordinate system lies in controlling the rigid motion of the “solid phase”, as well as in the prediction of the simultaneous phase transition process. The discontinuous physical properties on both sides of the zone lead to problems at the solid–liquid interface, usually referred to as a mushy zone. The widely used volume-of-fluid (VOF) method can solve the problem of changing discontinuous physical properties by transforming them into a property gradient related to the phase volume in the mushy zone. However, gradient properties, such as density, can introduce macroscopic forces at the interface, which may cause deformation of the “solid phase” or even lead to solution errors.

The VOF method has the potential to be applied to numerical simulations as long as the physical properties of the “solid phase” can be controlled. It should be noted that the biggest difference between fluids and solids is whether they can withstand shear stress. If the “solid phase” can withstand shear stress like a real solid, the deformation of the “solid phase” can be controlled at the interface. However, macroscopic motion can also occur within the “solid phase” due to this feature. Therefore, in this study, a simplified numerical model for the GDMM process is proposed based on the Euler coordinate system. A fluid deformation yield stress equation based on the Bingham assumption was adopted to simulate the rigid motion of the “solid phase”. A numerical description of the physical properties of the mushy zone was obtained using the VOF method. The ability of the model to predict phase interface position, volume fraction, and the thickness of liquid film was verified using the referenced experiments. Its enhanced performance when analyzing the convection of the liquid phase and the macroscopic motion of the medium were also numerically analyzed. This model provides a numerical method possessing a simplified solution and setup, which may be useful for analyzing the combination of the GDMM process and LTES devices.

## 2. Materials and Methods

Due to the use of the Euler coordinate system, all media in the computational domain were assumed to be fluids. The domain for simulating the solid phase was also treated as a liquid (represented by “solid phase” hereafter). Due to the existence of density differences between the medias, a relative motion was present between the liquid phase and the “solid phase”. To prevent the deformation of the “solid phase”, the Bingham fluid assumption was used to simulate the rigid motion of the “solid phase”. The motions, the heat transfer and the flow fields of the medias were physically described. Meanwhile, as these processes involved instantaneous phase interface changes, the phase interface was tracked. To simplify the development of the model, some assumptions are proposed herein:The liquid phase flow is assumed to be Newtonian, laminar and incompressible. Therefore, pressure and flow terms in the momentum equation are not considered;The substance in the “solid phase” domain is assumed to be the Bingham fluid that can withstand a certain level of shear stress [19];There is a mass transfer in the mushy zone. The physical properties are expressed through the volume fraction of the liquid phase;The Boussinesq approximation is introduced by using the gravitational term to simulate the buoyancy effect of the pure liquid phase [20];The phase transition energy related to the transfer of solid–liquid mass in the mushy zone is added to the energy equation as a source term.

Considering these assumptions, the governing equation, fluid property equation and mushy zone mass transfer equations are described below.

### 2.1. Governing Equations

The continuity equation is as follows:(1)$$\frac{\partial \rho }{\partial t}+\nabla \cdot \left(\rho \vec{u}\right)=0$$

It should be noted that although it is assumed that the density of all pure substances remains constant, Equation (1) is used to describe the mass conservation in the mushy zone. For the meshes in the mushy zone, the density is defined as *ρ* = *αρ_l_* + (1 − *α*) *ρ_s_*, where *α* is the volume fraction of the liquid phase and is defined as:(2)$$\alpha =\frac{V_{L}}{V_{cell}}$$

The momentum equation is as follows:(3)$$\frac{\partial \vec{u}}{\partial t}+\nabla \cdot \left(\vec{u}\vec{u}\right)=-\frac{1}{\rho }\nabla p+\nu \nabla^{2}\vec{u}+\beta \rho g(T-T_{f})$$
where *T_f_* is the reference temperature for buoyancy flow. The second viscosity term is small and can be ignored in the model solutions [17]. The Bingham fluid is applied through the stress term in Equation (3), which limits the shear rate (or the deformation rate) of the solid phase by setting the shear stress. If the shear stress applied to the “solid phase” mesh does not exceed the threshold value, the fluid inside the mesh exhibits rigid motion. The boundary of the “solid phase” is transmitted to internal forces through stress terms, and the adjacent “solid phase” meshes also rely on this to transfer forces that behave like rigid motions. The meshes in the mushy zone are affected by the volume fraction *α*. This causes changes in the density, which can lead to the formation of forces generated by density differences with the surrounding meshes. Therefore, the macroscopic motion of the “solid phase” can be directly driven by the stress term.

The energy equation is as follows:(4)$$\frac{\partial h}{\partial t}+\nabla \cdot \left(\vec{u}h\right)=\nabla \cdot \left(\frac{\lambda }{\rho }\nabla T\right)-L\frac{\partial \alpha }{\partial t}$$
where *h* is the enthalpy; *λ* is the thermal conductivity of the PCM; and *L* is the latent heat of the PCM. The second term on the right side of Equation (4) is the change rate in latent heat generated by the solid–liquid phase transition within the meshes in the mushy zone, which depends on the additional definition of the mass transfer rate in Section 2.3. Furthermore, for liquid or solid meshes, there is no phase transition inside, so the latent heat term does not affect the energy equation.

### 2.2. “Solid Phase” Property Equation

In order to achieve the rigid motion of the “solid phase”, its physical properties need to be specifically defined. Under the action of gravity, the solid phase underwent continuous macroscopic motion. Simultaneously, a pressure difference was observed in the vertical distribution of the solid phase, while the fluid flow in the liquid phase exhibited Newtonian behavior. To address this issue, the Bingham fluid assumption was introduced, which displayed the fluid characteristics capable of withstanding shear stress. The relationship between the shear stress (*τ*) and shear rate (*γ*) of the Bingham fluid can be seen in Figure 1.

Bingham plastic fluid, as a form of Bingham fluid, exhibits rigid body motions under low stress. The fluid only begins to flow when the shear stress reaches a specific value, at which point the shear stress changes linearly with the shear rate. Bingham plastic fluids have intermediate properties between viscosity and rheological properties, commonly found in engineering materials. The shear stress of Bingham fluids can be described by the Herschel–Bulkley model as follows [22]:(5)$$\tau =\tau_{0}+\nu {\left(\gamma \right)}^{n},\;(n=1\;\mathrm{for}\;\mathrm{Bingham}\;\mathrm{plastic}\;\mathrm{fluid})$$
where *τ*_0_ is the yield stress, *ν* is the dynamic viscosity and *n* is the power law index, which is a measure of the deviation between Newtonian and non-Newtonian fluids. During the melting process, the fluid behavior of the “solid phase” changed from Binghamian to Newtonian. With the introduction of the volume fraction, the mushy zone can be regarded as a gradient-based plastic fluid, which also conforms to the rheological properties of a Bingham fluid.

The critical shear stress (*τ*_0_) in this model was calculated from the initial phase interface position and the solid–liquid density difference (about several hundred Pa). This means that the largest pressure difference in the calculation domain was used to limit the deformation of the “solid phase”. Subsequently, the shear stress of the Bingham fluid represented by Equation (5) can be incorporated into the shear stress term in momentum equations, thereby achieving continuous solutions for solid-phase coupling and the liquid–solid phase transition in the mushy zone.

### 2.3. Defined Mass Transfer Function

In this study, a self-defined function to constrain the mass transfer process of the solid–liquid phase transition is developed, as shown by Equation (6). The equation is based on the conservation between the energy of obtained heat capacity and the latent phase change in the mushy zone mesh. In other words, from the right side of the equation, it can be seen that the mass of all the heat absorbed by the mesh is converted to the phase -change energy of the mesh. Otherwise, heat will pass through the mesh and enter the interior of the solid phase. However, this is not consistent with the physical reality. The constant coefficient ω is a relaxation factor, which was set as 0.1 in this study via volume fraction validation in simulations [17]:(6)$$m_{ls}=\omega \cdot V_{s}(T_{s}-T_{sat})\rho_{s}(C_{s}/L),\;\mathrm{if}\;T_{s}\geq T_{SAT}$$
where *m_ls_* is the transferred mass at each time step, and *T_SAT_* is the phase-change temperature of the PCM.

The relationship between mass transfer and the volume fraction is crucial for solving the model. An iterative correction scheme is shown in Equation (7). From this perspective, during the transient solution process, the mass transfer at each time step changes first, followed by updating the volume fraction and the physical properties of all the regions (due to their correlation with volume fraction):(7)$$\alpha ’=\alpha +\frac{m_{ls}}{\rho_{s}V_{s}}$$

In summary, this model can simulate multiphase flow by solving individual momentum equations by handling volume fractions within mesh cells and can accurately describe the mushy zone.

### 2.4. Solution Method

The VOF method is a surface tracking technique that is often applied to Euler coordinate system models and is commonly used in mature commercial software. When using the VOF method, the volume fraction has already been calculated. Therefore, in this study, the commercial software ANSYS Fluent (Version 2022) was used to solve the developed model with the VOF model. According to the mature model tree of Fluent software, users can conveniently set and solve model parameters, thereby avoiding the complex programming work of solving control equations. The detailed solution process is shown in Figure 2.

## 3. Model Verification

### 3.1. Meshing Independence

To illustrate the influence of mesh number on the developed model, an axisymmetric geometric model was selected, which is shown in Figure 3. It has overall geometric dimensions approximately 50 times that of the liquid-film thickness, which allows the mesh to be refined at the liquid film. Because the solutions in this film may be sensitive to the size of the mesh, other zones were not refined and were set using the global average mesh size. The specific mesh sizes are shown in Table 1.

The melting condition and the phase properties that were used were consistent with case 1 in Section 3.3. The time step was set as 0.0005 s, consistent with the other calculation cases outlined hereafter. The change in the liquid-phase volume fraction within 10 s was used for the meshing independence analysis. Considering the magnitude of the computations, each time step was iterated 10 times, and each case mentioned above required approximately 200,000 iterations. The comparison results are shown in Figure 4. It can be seen that the results of cases 1–3 are very close. The total number of meshes between case 3 and case 4 show a difference of 63,000, but there is an obvious deviation in the prediction. This indicates that mesh independence begins to be reflected in the range between case 3 and case 4. Therefore, all the cases in this article adopted the mesh sizes of 0.02 mm for the global size and 0.01 mm for local refining.

### 3.2. Macroscopic Settling Velocity of “Solid Phase”

To illustrate the macroscopic settling performance of the “solid phase” with the Bingham fluid assumption, vector velocity distributions near the liquid film with and without the Bingham fluid assumption are shown in Figure 5. The mass transfer was not considered, and the right-angled area at the bottom of the “solid phase” was emphasized because, in this area, the liquid phase influences the direction of the flow, which may cause deformation of the “solid phase”. In our simulations, calcium dichloride hexahydrate (CaCl_2_·6H_2_O) was used as the PCM. The solid–liquid density difference was 168 kg m^−3^, which can drive the macroscopic settling of the “solid phase”. It can be seen in Figure 5 that the Bingham fluid assumption (with a threshold shear stress of 100 Pa) can effectively control the flow deformation of the “solid phase”. Furthermore, the selected threshold shear stress is larger than the pressure difference at the solid–liquid interface, which must be considered and pre-estimated in simulations.

### 3.3. Verification Case 1: Vertical Cylinder Melting

#### 3.3.1. Referenced Experimental Data

Pan et al. [15] experimentally investigated the melting process of calcium dichloride hexahydrate (CaCl_2_·6H_2_O) in a vertical cylindrical container. The detailed thermophysical properties of the PCM are shown in Table 2. The PCM was placed in a sealed glass tube (with some gas reserved) and vertically submerged in a constant temperature water bath for the melting. The liquid fraction during melting was recorded and extracted from images.

#### 3.3.2. Model Development

The calculation domain was established based on the experimental setup, which is shown in Figure 6. The two-dimensional computational domain was established in an axisymmetric form, corresponding to the working condition with 10 g of PCM. The computational domain includes three fluid regions: air, liquid phase, and “solid phase” at the initial point. Considering that the total volume of the PCM changes during the solid–liquid phase transition process, the air region was used as a buffer zone for fluid expansion. The bottom and side walls were assigned constant wall temperature conditions. In addition, to ensure the normal startup of the model, a thin liquid film region with a thickness of 100 μm was set up (without this setting, the model would not be able to start the calculations).

Based on the mesh independence analysis results, the total number of meshes was about 300,000 in simulations. It is notable that, due to the influence of thermal conductivity between the liquid and “solid” phases, the contour line of the phase-transition temperature *T_SAT_* can easily exceed the phase interface and enter the interior of the “solid phase”. Therefore, the thermal conductivity of the “solid phase” was set to 0.0 W m^−1^ K^−1^. This means that there is no heat conduction inside the “solid phase”, which is also similar to the physical reality.

#### 3.3.3. Verification Results

Figure 7 shows the comparison of the simulated results and the experimental measurements reported by Pan et al. [15] on the variation in the liquid-phase volume fraction over time at the driving-temperature difference of 10 °C and 20 °C. By using the root mean squared error (RMSE) method (shown in Equation (8)), the deviations between the experimental and simulated data were obtained as 7.68% (ΔT = 20 °C) and 10.88% (ΔT = 10 °C).
(8)$$RMSE=\sqrt{\frac{\sum_{i=1}^{N}{\left(E_{i}-{\hat{E}}_{i}\right)}^{2}}{N}}\times 100\%$$

Furthermore, from both experiments and simulations, it was determined that the variation in the volume fraction slowed down in the later stage of melting, which also indicates that the strengthening effect of GDMM on the phase transition became smaller compared to the increase in thermal resistance.

Figure 8 shows the comparison of the melting patterns from the experimental images and simulations with the driving-temperature difference of 20 °C. The compared liquid–solid phase change interfaces with similar shapes are highlighted by using the dash rectangles and circles. It can be seen that a convex and nonuniform phase boundary appears at the top of the PCM solid phase, which may relate to the fluidity of the liquid phase. A similar phenomenon was also observed in the simulation. The predictions of both the volume fraction and phase distribution demonstrate the consistency between the model and the experiments.

### 3.4. Verification Case 2: Thin Liquid-Film Thickness

#### 3.4.1. Referenced Experimental Data

Hu et al. [23] used laser interferometry to measure thin liquid-film thickness during PCM melting in a horizontal heated plate with constant temperature. A circular tube mold container with a diameter of 50 mm and a height of 20 mm was utilized. The liquid film-thickness variations were measured by using a high-speed camera through laser interferometry. It should be noted that the study by Hu et al. obtained, for the first time, dynamic and accurate measurements of thin liquid-film thickness. The PCM properties of the experiment are shown in Table 3.

#### 3.4.2. Model Development

To verify the dynamic perdition performance of the proposed model for thin liquid-film thickness, a two-dimensional axisymmetric geometry model and mesh were developed, as shown in Figure 9. The total mesh number was 380,000. The constant wall temperature boundary was used. In addition, to ensure the startup of the model, an initial thin liquid-film thickness below 100 μm was set.

#### 3.4.3. Verification Result

The measurements and simulation values of the thickness of a thin liquid film are related to the width of the mushy zone. The mushy zone is composed of a partial liquid and partial solid phase. To the best of our knowledge, there is currently no clear method for determining the simulated value of thin liquid-film thickness. Therefore, according to both our simulation results and the experiments of Hu et al., the simulated thickness values were determined by measuring the middle position of the mushy zone, i.e., the 50% volume fraction. The detailed determination method can be seen in Figure 10.

Figure 11 shows the comparison results of the thin liquid-film thickness within 5 s (the most obvious variation stage of the thickness) in the melting process at driving temperature differences of 10 °C and 30 °C. The initial heights of the PCM solid phase of 20 mm and 40 mm were also considered. It can be seen that the thickness variation trend and the magnitude were all consistent with the experiment. It is worth noting that the initial time point of the simulation was corrected through the predicted initial thickness with the experimental value. Therefore, the thickness values at subsequent moments could be compared. At the beginning of melting, the thickness of the liquid film increased rapidly, and after 2 s, the rate rapidly slowed down. Moreover, a lower driving-temperature difference and a higher initial height of the PCM solid phase resulted in a thinner liquid film. This change trend can also be found in the literature [23], which proves that the model has good consistency.

## 4. Numerical Analysis and Results

### 4.1. Convective Heat Transfer within the Thin Liquid Film

In order to estimate the convective enhancement ability of the GDMM process on the liquid phase, the Nusselt number (*Nu*) was used to indicate the ratio of convective heat to conduction heat. The heated bottom wall was treated as the main object of analysis, corresponding to the thin liquid film, as this is where the flow enhancement began. Due to the continuous settling and melting of the “solid phase”, the liquid phase near the heated bottom wall is forced to flow to other areas, which starts to enhance the overall fluidity of the liquid phase. However, it should be noted that, due to the small thickness of the thin liquid film, thermal conductivity within this area still dominates the heat transfer process. Therefore, to illustrate the degree of flow enhancement, the averaged Nusselt numbers of the heated bottom wall were extracted from the simulations. The GDMM Nusselt number was calculated based on Equation (9):(9)$$Nu_{\mathrm{con}}=\frac{\bar{U}d}{\lambda }$$
where, $\bar{U}$ is the average convective heat transfer coefficient of the heated bottom wall obtained from the simulations.

Meanwhile, the comparison between the averaged Nusselt number and other convective intensities was also considered. The natural convective Nusselt number was analogously derived from horizontal cavities. It is calculated by using the height of the rectangular calculation domain as the characteristic length, which represents the possible maximum natural convection strength with a heated bottom wall in a horizontal cavity. Based on this assumption, the *Gr* is within the range of 1.0 × 10^4^ ≤ *Gr* ≤ 4.6 × 10^5^ and the natural convective Nusselt number can be calculated by Equation (11):(10)$$Gr=\frac{g\beta \Delta Td^{3}}{{\left(\frac{\nu }{\rho }\right)}^{2}}$$
(11)$$Nu_{\mathrm{nat}}=0.212{\left(Gr\cdot Pr\right)}^{\frac{1}{4}}$$
where the Prandtl number is calculated as:(12)$$Pr=\frac{C\cdot \nu }{\lambda }$$

Based on the data of the verified case 1, Figure 12 shows the comparison results of *Nu* with two different driving temperature differences for natural convection and the GDMM process. It can be seen that the GDMM Nusselt number of the heated bottom wall increases with an increase in the liquid phase volume fraction. This indicates that, as the melting process continues, more flow areas for the liquid phase are produced. It also conversely shows that the liquid phase fluidity in the thin liquid film area is weak. However, this does not affect the high heat transfer rate within the thin liquid film due to its extremely small thickness. Compared with the natural convection Nusselt number, the GDMM Nusselt number reaches a larger value before the halfway point of the melting process. This indicates that the convective intensity during the GDMM process is greater than that of the natural convection of the pure liquid phase. This also means that not only does the thin liquid film maintain a high heat transfer rate, but also that the flow enhancement of the GDMM process can also begin to play a role in the middle and later stages of melting. Therefore, this process can significantly improve the heat transfer rate and shorten the melting time.

### 4.2. The Settling Velocity Variation with Pressure Difference

The settling velocity of the “solid phase” during the GDMM process can serve as an indicator of the vertical phase transition rate. The larger the velocity, the faster the melting rate that can be achieved. This study extracts the velocity of the top point of the “solid phase” on the axis of symmetry and the pressure difference between the top point and the bottom point on the axis of symmetry. The variations in these two parameters are shown in Figure 13. It can be seen that there is a roughly linear relationship between the settling velocity and the pressure difference. This can be explained by Equation (3), in which the influence of the temperature-driven buoyancy term on the macroscopic motion can be ignored for the macroscopic settlement of the “solid phase”. The settling velocity is only related to the shear stress caused by the density difference between the liquid and solid phases. Therefore, a linear relationship between the settling velocity and the pressure difference is formed. In a study by Fu et al. [24], additional pressure was applied using a piston movement on the top of the solid phase. The experimental results demonstrated that there is a linear relationship between the applied pressure and the vertical movement velocity of the piston. This also demonstrates the reliability of the proposed model.

## 5. Conclusions

In this study, a Eulerian numerical model for simulating GDMM process was established. The Bingham fluid assumption was introduced to constrain the fluidity of the “solid phase”. Then, the VOF method and self-defined mass transfer function were used to numerically solve the model. The reliability of this model was verified by the referenced data. Our main conclusions are as follows:Based on the Bingham fluid assumption, it can be seen in the simulation results that the flow deformation of the “solid phase” can be controlled by setting the threshold value to the shear stress term. The model is used to predict the melting processes in closed cylindrical containers. The prediction error of the transient volume fraction is less than 10.88%. The predicted melting pattern for the “solid phase” indicates good consistency with the experiment. Selection criteria for the simulated liquid-film thickness were proposed, for which the predictions provided rational results;The estimation of the convective enhancement ability of the GDMM process on the liquid phase shows that the GDMM process can accomplish a flow-strengthening effect in the middle and later stages of melting. This explains the mechanism of the high melting rate. In addition, a linear relationship between the settling velocity and the pressure difference of the “solid phase” is observed, which is consistent with the results of Fu et al., obtained under external pressure conditions.

The proposed model has advantages in simplifying the solution procedures. The next step will be to conduct simulation research in conditions, including more complex geometries and analyses, that may be required for the optimization of real devices.

## Figures and Tables

**Figure 1 entropy-26-00175-f001:**
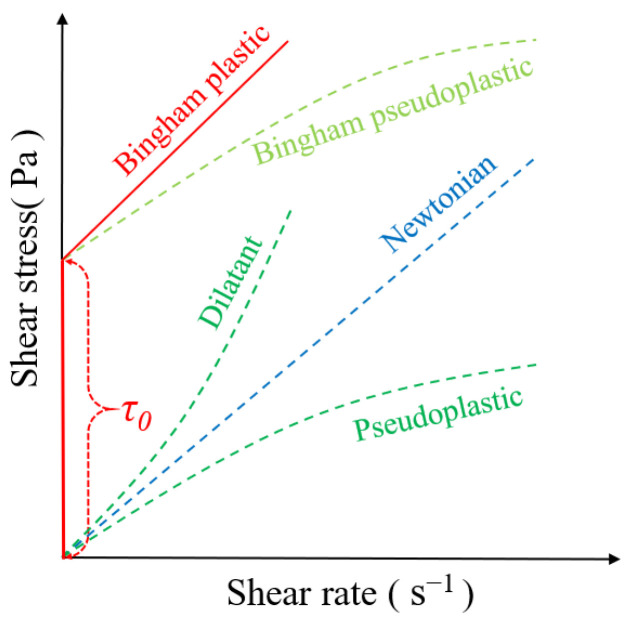
Shear stress vs. shear rate for different kinds of fluids [21].

**Figure 2 entropy-26-00175-f002:**
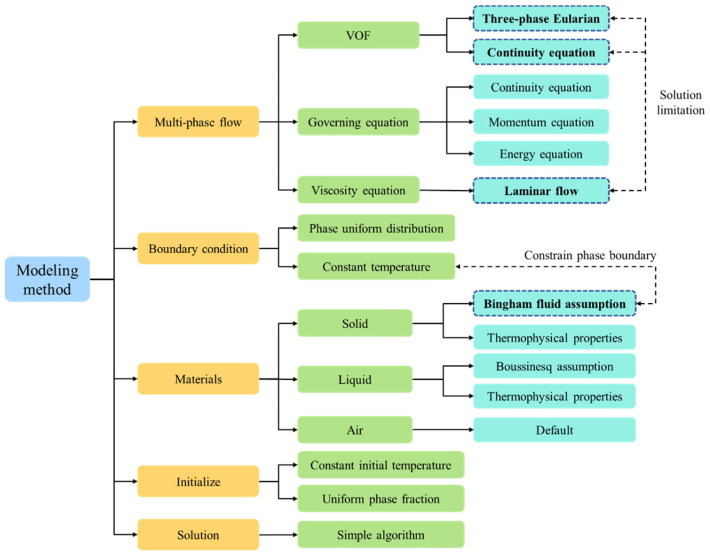
Solving framework of the proposed model.

**Figure 3 entropy-26-00175-f003:**
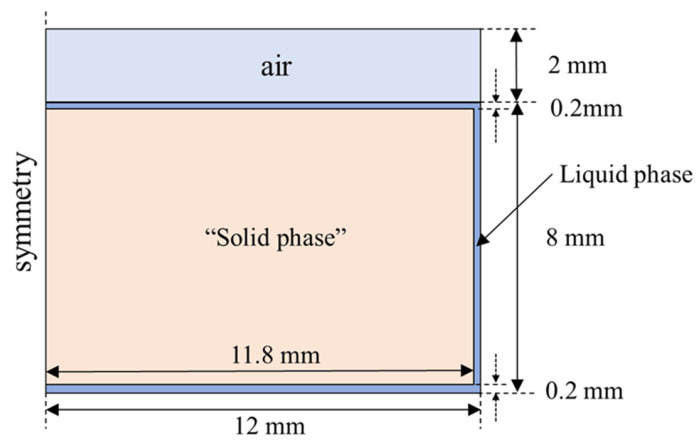
Geometry model for mesh size independence analysis.

**Figure 4 entropy-26-00175-f004:**
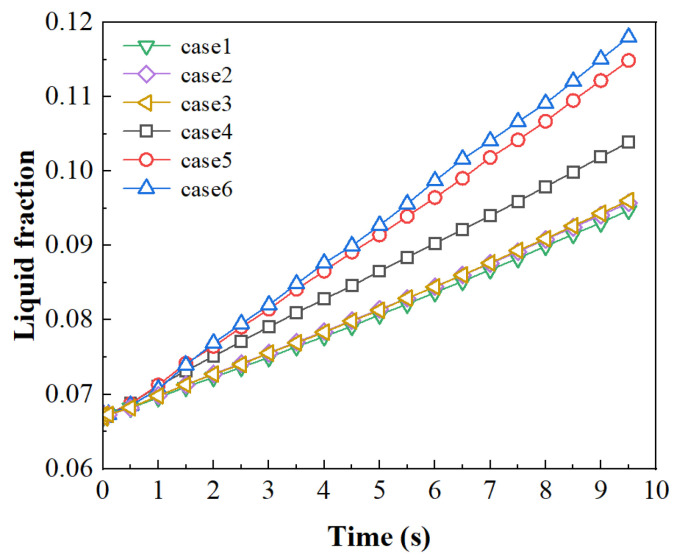
Variation of volume fractions with different mesh sizes.

**Figure 5 entropy-26-00175-f005:**
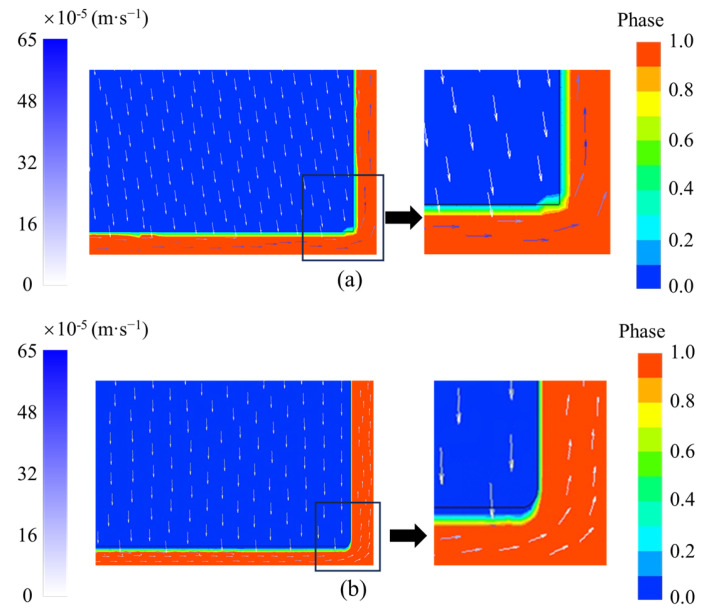
Vector velocity distribution nearby the liquid film at 10 s: (**a**) without Bingham assumption; and (**b**) with Bingham assumption.

**Figure 6 entropy-26-00175-f006:**
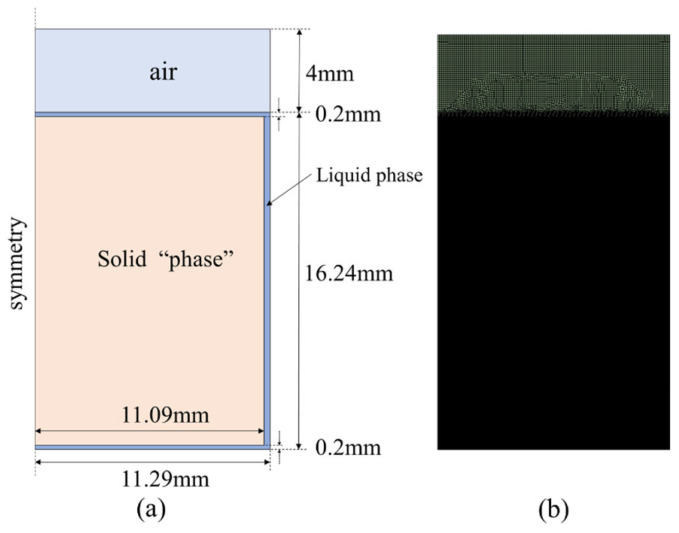
Calculation domain for case 1: (**a**) dimension; and (**b**) meshing.

**Figure 7 entropy-26-00175-f007:**
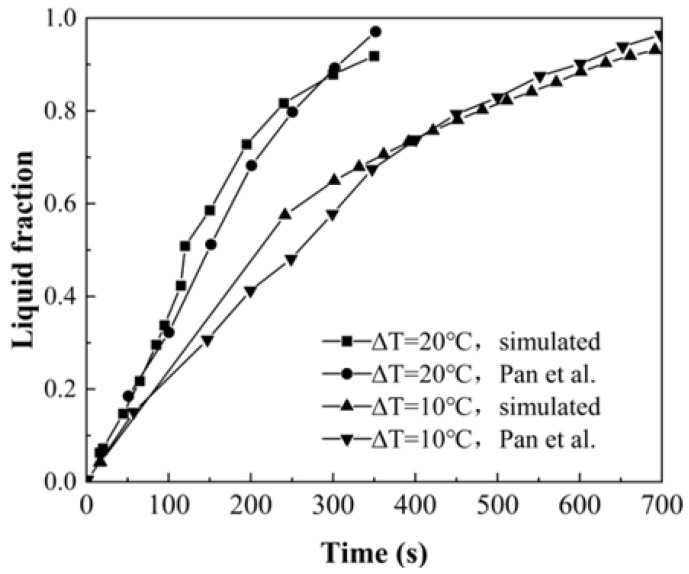
Comparison of experimental and simulated liquid fractions.

**Figure 8 entropy-26-00175-f008:**
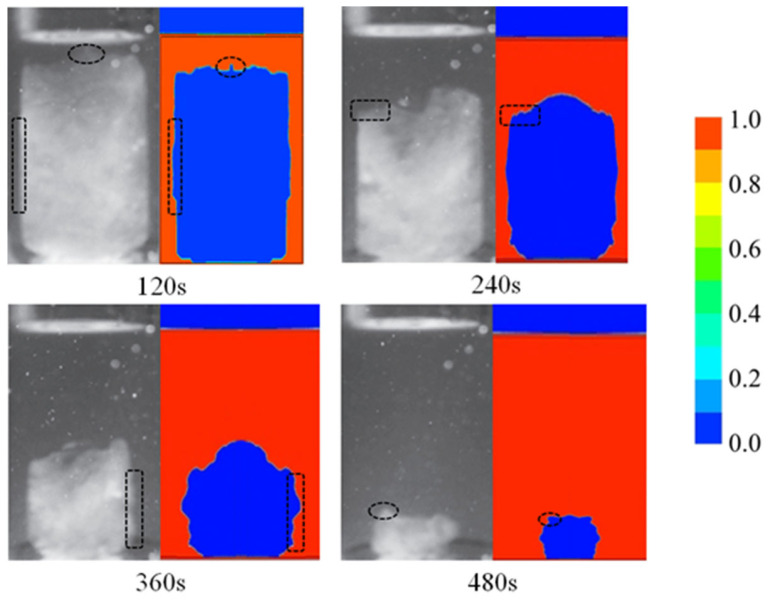
Experimental and simulation comparison of phase distribution.

**Figure 9 entropy-26-00175-f009:**
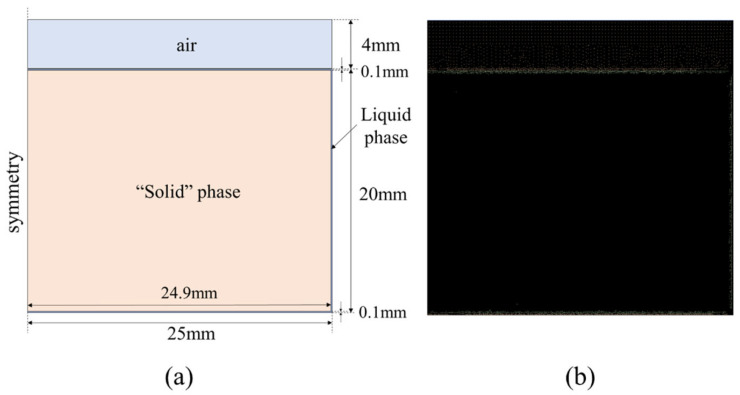
Calculation domain for case 2: (**a**) dimension; and (**b**) meshing.

**Figure 10 entropy-26-00175-f010:**
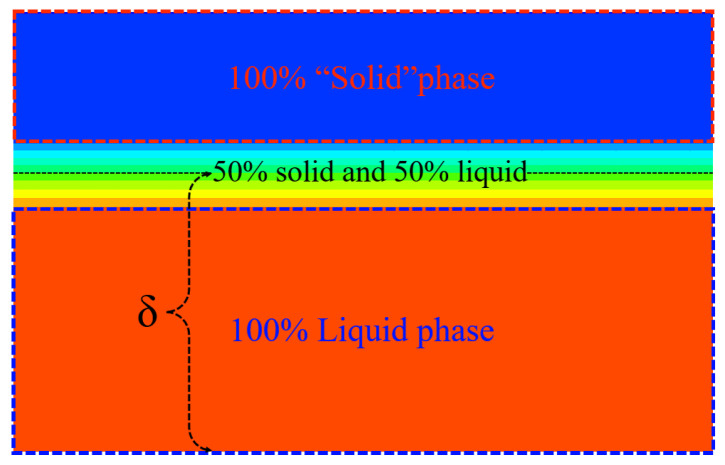
Measurement method of thin liquid-film thickness in the simulations.

**Figure 11 entropy-26-00175-f011:**
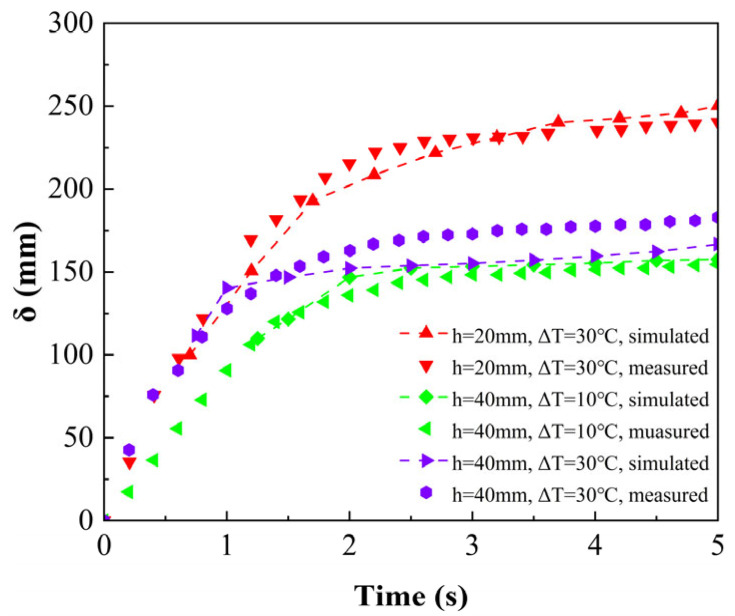
Transient variations of the thin liquid-film thickness: simulated vs. measured.

**Figure 12 entropy-26-00175-f012:**
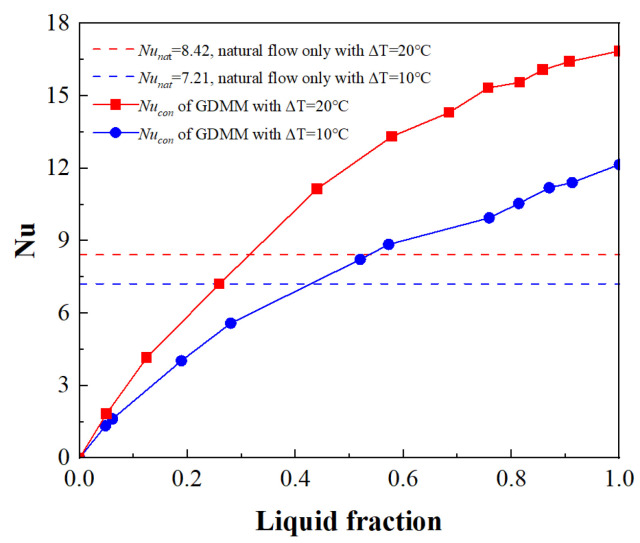
Nusselt number comparison: natural vs. GDMM convection.

**Figure 13 entropy-26-00175-f013:**
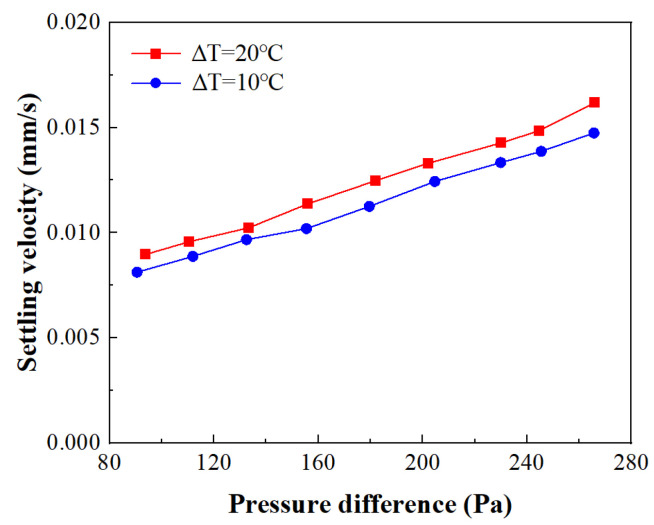
Variation of settling velocity with vertical pressure difference.

**Table 1 entropy-26-00175-t001:** Mesh parameters of geometry model.

Case	Average Mesh Size in the Solid Phase Region (mm)	Refined Mesh Size in the Liquid Phase Region (mm)	Total Number of Mesh Cells
1	0.02	0.006	384,680
2	0.02	0.008	323,596
3	0.02	0.01	295,984
4	0.02	0.02	232,976
5	0.03	0.03	108,466
6	0.04	0.04	64,923

**Table 2 entropy-26-00175-t002:** Thermophysical properties of PCM.

Properties	Values
Melting temperature	29 (°C)
Density (solid/liquid)	1706/1538 (kg m^−3^)
Thermal conductivity (solid/liquid)	1.09/0.546 (W m^−1^ K^−1^)
Specific heat (solid/liquid)	2060/2230 (J kg^−1^ K^−1^)
Molecular weight	219.08
Coefficient of thermal expansion	0.0005 (K^−1^)
Latent heat	170 (kJ kg^−1^)
Dynamic viscosity	0.01 (Pa s)

**Table 3 entropy-26-00175-t003:** Thermophysical properties of 1-tetradecanol.

Properties	Values
Melting temperature	39 (°C)
Density (solid/liquid)	869/802.70 (kg m^−3^)
Thermal conductivity (solid/liquid)	0.85/0.35 (W m^−1^ K^−1^)
Specific heat (solid/liquid)	376.66/704.51 (J kg^−1^ K^−1^)
Molecular weight	214.39
Coefficient of thermal expansion	0.0032 (K^−1^)

## Data Availability

The detailed data involved in this study can be prepared and provided to reviewers and readers as required.

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
