# Peer review of "A Eulerian Numerical Model to Predict the Enhancement Effect of the Gravity-Driven Motion Melting Process for Latent Thermal Energy Storage"

_entropy, 2024, doi:10.3390/e26020175_

Round 1

Reviewer 1 Report

Comments and Suggestions for Authors

This article deals with a numerical model for the melting behavior of phase-change materials, particularly when gravity and natural convection increase heat transfer. This phenomenon is referred to here as Gravity-Driven-Motion-Melting (GDMM), and usually as Close-Contact-Melting (CCM). The article is interesting, but I have several comments, suggestions and questions for the authors:

- What's the difference between GDMM and CCM? If there isn't one, why change the name? Even the references you cite when talking about GDMM only use the term CCM.

- Reference 5, used to prove IRENA's claim that energy storage will increase, is not adequate. The reference used has a different purpose. Please cite an appropriate report from this agency.

- Could you explain clearly what motivates GDMM/CCM in the introduction? I mean, what are the criteria that lead to this effect? Use references to explain your assertions. An illustration of the phenomenon might be a plus.

- CCM has been well known for decades now. Some articles have already highlighted a 7-fold increase in heat transfer with CCM in 1986. I think readers need a better summary of what we know about the phenomenon, and what's missing.

- The literature review could be improved by avoiding general descriptions of other work, focusing more on results and conclusions and adding further references. What about the work of Shockner et al (2021), Rozenfeld et al (2015) and Oskouei et al (2024)? Similarly, the work of Fu et al (ref 24) could be cited in the introduction to give recent results on this phenomenon. (I'm not asking you to add these references, but simply to take them into account, and to do more literature review to find other relevant works).

- reference 21 is used to prove an equation, but it is a software reference. Please use an appropriate scientific reference

- Would it be possible to supply the model with the article? I think that when a new model is described and it is claimed that it could be useful to the scientific community, it should be shared with the article or on a dedicated platform (code, files...).

- p.9,l.290: "It can be seen that there is a convex and nonuniform phase boundary appears at the top of the PCM solid phase, which may relate to the fluidity of the liquid phase." Isn't it possible to confirm this hypothesis with a dedicated simulation (with and without the fluidity)?

- In their article, Pan et al (ref 22) have also developed a model. Could you explain in the introduction how yours is better or not?

- In their paper, Pan et al (ref 22) have given experimental and numerical shapes of the solid for different times. Please consider doing the same.

-Section 4.1: Start by explaining the purpose of this section, then describe the Nusselt numbers. If I understand correctly, you are estimating an effective Nusselt number that includes CCM/GDMM, and comparing it to natural convection only. If so, please rephrase to make it clearer. In any case, would it be possible to give another figure that shows the differences in the propagation of the melting front?

- section 4.2: I'm not sure I understood the purpose of this section.

- I don't think the novelty of this work is clearly explained. Is the model just simpler? To what extent is it faster yet simpler? Or to what extent is it more accurate than another?

Also:

- Figure 5: better separate a and b to see the borders of each image. Wouldn't it be more interesting to zoom in on the lower right-hand corners?

- Figure 12: please reword the legend. Is it "natural convection only" and "effective convection thanks to CCM"?

Comments on the Quality of English Language

English could be much improved.

- For example, starting a sentence with "wherein" is uncommon or incorrect. Please use a digital tool to improve your English, or ask a native English speaker to check and correct the language.

- "The GDMM process has a given feature that involved in fluid solid coupling and liquid-solid phase transition simultaneously". This is an example of a sentence I do not understand. Please rephrase.

-  "It can be seen that the results of case 1-3 are seems clos". This is an another example of wrong grammar.

- When you talk about "settling" of the solid phase, do you mean "sinking"?

- The conclusion should be rewritten. Too many grammatical errors

Author Response

Thanks for the reviewer's suggestions. The point-by-point response can be found in the attachment.

Reviewer 2 Report

Comments and Suggestions for Authors

This is an interesting paper. I am not convinced that all papers cited in the introduction are directly relevant to the study.

Methodology is ok but not sufficiently detailed. It would be good to revisit this section to make sure it is clear and that readers are able to replicate the simulation based on data provided. Specifically it would be good to explain which value were assumed (for example, relaxation factor of 0.1 on p5) and why. It would be welcome to explain "The “solid phase” domain is assumed to be a Bingham fluid that can withstand shear stress [18]." on p3. And also on p10 "It should be noted that the study by Hu et al. has, for the first time, obtained dynamic and accurate measurements of the thin liquid film thickness."; how would results be affected if different values were used?

Author Response

Thanks for the reviewer's comments. We have up load the point-by-point response as the attachment file.

Round 2

Reviewer 1 Report

Comments and Suggestions for Authors

Thanks for improving the article. I have no further comments.

Comments on the Quality of English Language

There are still some grammatical errors in the new paragraphs. Please consider using an intelligent english corrector.

For instance: " there are extremely thin liquid films (about hundred micrometers) can be formed between heated wall and solid phase"

It could be "Extremely thin liquid films (of the order of a hundred micrometers) can form between the heated wall and the solid phase".

Also: "The bottom heated wall is treated as the mainly analysis object, which is correspond to the thin liquid layer"

"The heated bottom wall is treated as the main object of analysis, corresponding to the thin layer of liquid." ?

Author Response

Yes. The detailed response has been upload as an attachment file.
